# Rapamycin-Induced Feedback Activation of eIF4E-EIF4A Dependent mRNA Translation in Pancreatic Cancer

**DOI:** 10.3390/cancers15051444

**Published:** 2023-02-24

**Authors:** Trang Uyen Nguyen, Harrison Hector, Eric Nels Pederson, Jianan Lin, Zhengqing Ouyang, Hans-Guido Wendel, Kamini Singh

**Affiliations:** 1Department of Molecular Pharmacology, Albert Einstein College of Medicine, Albert Einstein Cancer Center, Bronx, NY 10461, USA; 2Department of Biostatistics and Epidemiology, School of Public Health and Health Sciences, University of Massachusetts Amherst, Amherst, MA 01003, USA; 3Cancer Biology and Genetics, Memorial Sloan Kettering Cancer Center, New York, NY 10065, USA

**Keywords:** mTOR, eIF4E, eIF4A, p70-RSK1, ribosome footprinting, CR-1-31B

## Abstract

**Simple Summary:**

Pancreatic cancer is aggressive cancer with a low survival rate due to the lack of detection, effective treatment, and development of therapeutic resistance. New treatments and mechanistic details of therapeutic resistance are urgently needed. In this study, we explored the effect of inhibiting protein synthesis and its role in inducing feedback mechanisms that may impact the therapeutic response. We show that Rapamycin (sirolimus) treatment inhibited the synthesis of proteins required for cancer cell growth. Interestingly, rapamycin treatment induced the synthesis of proteins that lead to reactivation of the key kinases, and this limited the anti-tumor effect of rapamycin. We further show that the combination of rapamycin with the small molecule inhibitor CR-1-31-B increases the efficacy of rapamycin. Our study establishes the feedback mechanism induced by rapamycin and new therapeutic combinations that can be further developed as therapeutics for pancreatic cancer.

**Abstract:**

Pancreatic cancer cells adapt molecular mechanisms to activate the protein synthesis to support tumor growth. This study reports the mTOR inhibitor rapamycin’s specific and genome-wide effect on mRNA translation. Using ribosome footprinting in pancreatic cancer cells that lack the expression of 4EBP1, we establish the effect of mTOR-S6-dependent mRNAs translation. Rapamycin inhibits the translation of a subset of mRNAs including p70-S6K and proteins involved in the cell cycle and cancer cell growth. In addition, we identify translation programs that are activated following mTOR inhibition. Interestingly, rapamycin treatment results in the translational activation of kinases that are involved in mTOR signaling such as p90-RSK1. We further show that phospho-AKT1 and phospho-eIF4E are upregulated following mTOR inhibition suggesting a feedback activation of translation by rapamycin. Next, targeting eIF4E and eIF4A-dependent translation by using specific eIF4A inhibitors in combination with rapamycin shows significant growth inhibition in pancreatic cancer cells. In short, we establish the specific effect of mTOR-S6 on translation in cells lacking 4EBP1 and show that mTOR inhibition leads to feedback activation of translation via AKT-RSK1-eIF4E signals. Therefore, targeting translation downstream of mTOR presents a more efficient therapeutic strategy in pancreatic cancer.

## 1. Introduction

Pancreatic cancer (PDAC) is driven by mutant KRAS that feeds to PI3K-AKT-mTOR signaling to support anabolic pathways and cancer cell growth [1,2]. The mammalian rapamycin complex 1 (mTORC1) target is a central eukaryotic signaling complex that coordinates metabolism and cell growth [3,4]. Due to the role of mTORC1 in the initiation of protein translation and ribosome biogenesis, altered regulation of the activity of this complex has been implicated in numerous cancers, including PDAC [4]. mTORC1-dependent activation of S6K results in phosphorylation of S6 [5]. S6K also phosphorylates several other targets which are associated with enhanced translation such as the eukaryotic translation initiation factors 4B and 4F (eIF4B and eIF4F, respectively) and the eukaryotic elongation factor 2 kinase (eEF2K) [6,7]. Despite many targets of S6K being identified, inactivation of S6K has not been shown to affect global translation rates [8], though the prevention of phosphorylation of another mTOR target, 4E-BP1, does indeed substantially decrease the translation, specifically of mRNA containing 5′ terminal oligopyrimidine (5′-TOP) motifs [9,10]. Phosphorylation of 4E-BP1 results in dissociation from eIF4E, which may increase the affinity of the eIF4F complex to these 5′-TOP and TOP-like mRNA motifs [10,11]. Furthermore, the knockdown of 4EBP1/2 has been shown to greatly reduce the effect of mTOR inhibitor Torin1 on the TOP and TOP-like motif-dependent translation [10]. Due to the pathway convergence, the activity of mTORC1 can be affected by changes in the activity of many upstream proteins, which is often implicated in cancer, leading to increased activity of both S6K and 4E-BP1 [12,13].

The mTORC1 complex plays a central role in cancer development meaning its inhibition has been the target of numerous therapeutic studies [14,15]. Rapamycin was the first mTOR inhibitor to be identified, initially being proposed as an antifungal antibiotic, but has since been studied in the context of cancer treatment [16]. Despite this, targeting mTOR has not been effective in PDAC for many reasons including the fact that 4EBP1 is often lost in PDAC [17]. One explanation for this may be that mTOR specifically regulates the translation of ribosomal proteins and mRNAs containing TOP motifs in cells that have intact S6 and 4EBP1 signaling [10]. Another contributing factor could be that the phosphorylation of S6 alters the translation of shorter coding sequences (CDS) to a greater extent than longer CDS [18]. While many mTOR inhibitors have been developed and extensively studied in the context of 4EBP1 and cap-dependent translation, the contribution of S6 on translation is still poorly understood in pancreatic cancer [19,20,21]. We, therefore, set out to determine the translational changes induced by rapamycin in pancreatic cancer cells.

## 2. Results

### 2.1. Ribosome Footprinting Identifies Translational Targets of mTOR and RPS6 (S6)

Human pancreatic cancer cells PANC-1, MiaPaca2, and PANC10.05 show growth inhibition following rapamycin treatment for three days, MiaPaca2 showing higher sensitivity compared to PANC-1 and PANC10.05 (Figure 1A and Appendix A). Next, we measured the effect of rapamycin on global mRNA translation using AHA labeling. We observed up to 10% inhibition at 1 h and ~50% inhibition in global mRNA translation at 4 h following rapamycin treatment (Figure 1B). In correspondence with these findings, in a long-term colony formation assay, we observe growth inhibition following rapamycin treatment (Figure 1C,D and Appendix A). To characterize the effect of rapamycin on genome-wide translation, we utilized the technology of ribosome footprinting. Briefly, we treated PANC-1 cells with DMSO (*n* = 3) or rapamycin (50 nM; for 1 h; *n* = 3) and then deep sequenced the total RNA and ribosome-protected fragment (RF) RNA (Figure 1E). Quality control analysis of RNA and RF replicates showed significant correlations among the replicates with a Pearson coefficient >0.99 and >0.97, respectively (Appendix A). Read mapping to ribosomal RNAs, non-coding RNAs, library linkers, and incomplete alignments were removed from the analysis. Most of the remaining reads range from 25 to 35 nucleotides in length and map to protein-coding genes (Appendix A). The total number of RF reads mapped to exons was 2.7 million in DMSO and 2.3 million in rapamycin-treated samples. This corresponds to 20,213 protein-coding genes. We used the RiboDiff statistical framework to isolate the effect on mRNA translation [2]. 

With a very stringent statistical cut-off at q < 0.01 (FDR < 1%) and q < 0.05 (FDR < 5%), we identified 473 and 861 mRNAs whose translation was significantly repressed (TE down) (Figure 1F). We also detect a set of mRNAs showing a relative increase in ribosome occupancy (TE up *n* = 588; q < 0.01 and *n* = 954; q < 0.05) (Figure 1F). A full list of genes differentially affected by rapamycin is provided in Appendix A. Hence, we identified the subset of mRNA whose translation is differentially regulated upon mTOR inhibition.

### 2.2. Rapamycin Inhibits the Translation of Genes Involved in the Cell Cycle and Cancer Cell Growth

We noticed that the TE down mRNAs showed a significant reduction in ribosome coverage through the 5′UTR and CDS of the transcripts suggesting that mTOR inhibition results in an overall reduction in the ribosomal occupancy on the affected transcripts (Figure 2A TE down q < 0.01 and Appendix A TE down q < 0.05). Next, we performed GSEA of the TE down genes (*n* = 861; q < 0.05) and observed KEGG pathways enrichment for cysteine and methionine metabolism, proteasome, peroxisome, cell cycle, and pathways in cancer (Appendix A). We also observed Hallmark pathway enrichment for peroxisome, androgen response, UV response, mTORC1 signaling, MYC, and E2F targets (Appendix A). mRNA transcripts whose translation is downregulated by rapamycin are ranked by q value (q < 0.01 and q < 0.05 cut-off) showing many genes involved in the cell cycle and cancer cell growth (Figure 2B). Notably, rapamycin treatment reduced the translation of genes involved in the cell cycle, Fc Gamma receptor-mediated phagocytosis, cysteine, methionine metabolism, and pathways in cancer (Figure 2C,D). This includes proteins such as MDM2, RPS6KB1 (p70 S6K), RPS6KB2 (p70 S6K), GOT2, SMAD4, and CDK2 (Figure 2B–D). Ribosome coverage showed a significant reduction in examples from the TE down groups, such as EPCAM, SMAD4, and MDM2 while the total RNA of these genes remained unaffected following rapamycin treatment (Figure 2E–G, Appendix A). Next, we validated the effect of translational change on the protein expression of RPS6KB1 and RPS6KB2. The total protein decreased following rapamycin treatment on PANC-1 cells while the total RNA remained unchanged (Figure 2H and Appendix A). Full immunoblots are shown in Appendix A. These data suggest that rapamycin inhibits the translation of genes involved in cancer cell growth including p70 S6K, a downstream effector of mTOR signaling. Rapamycin is reported to affect the translation of TOP, TOP-like, and PRTE motifs [10,22]. In line with these findings, we observed the enrichment of TOP and PRTE motifs in the 5′UTR of TE down mRNAs (Appendix A). Additionally, we found that Rapamycin inhibited the TOP-dependent translation activity in PANC1 cells (Appendix A).

### 2.3. Rapamycin Induces Translation of a Subset of mRNAs

Interestingly, we observed that a greater number of mRNAs are translationally upregulated (*n* = 588) compared to TE downregulated (*n* = 473) suggesting that mTOR inhibition in PANC-1 cells has a profound effect on the upregulation of translation (Figure 1E). Next, we analyzed the ribosomal coverage on the subset of TE-up mRNAs and observed increased ribosomal occupancy throughout the length of the transcripts (Figure 3A and Appendix A). GSEA shows enrichment of pathways related to PI3K-AKT-mTOR, IL6-STAT-JAK, androgen, TNF-NFkB, UV response, and IL2-STAT5 signaling (Appendix A). Transcripts whose translation is upregulated by rapamycin are shown as ranked by q value (q < 0.01 and q < 0.05 cut-off) (Figure 3B). Genes involved in PI3K-AKT-mTOR, TNFa signaling, IL2-STAT5, and mTORC1 signaling are translationally upregulated following rapamycin treatment in PANC-1 cells (Figure 3C). Ribosomal coverage on candidate genes, e.g., RPS6KA1 (p90 RSK1), PDK1, and MKNK2shows upregulation of ribosome throughout the transcript length except for 3′UTR (Figure 3D–F). After rapamycin treatment, we validated the upregulation of total protein for RPS6KA1 (p90 RSK1) and STAT5A (Figure 3G,H). Full immunoblots are shown in Appendix A. Consistently, the total RNA for these candidate genes remained unaffected suggesting that the protein is upregulated by increased translation (Appendix A).

### 2.4. Rapamycin Activates Translation through Feedback Activation of AKT1 and eIF4E

Rapamycin treatment has previously been shown to activate phospho-AKT1 and eIF4E [23,24,25]. In PANC-1 cells, we observe that phospho-AKT1 is activated within 5 min and stays activated at 60 min following rapamycin treatment while phospho-S6 is inhibited as early as 10 min and remains inhibited until 60 min (Figure 4A).Additionally, we found that phospho-eIF4E is activated as early as 5 min and remains activated at 60 min following rapamycin treatment in PANC-1 cells (Figure 4B). Phospho-p90-RSK1 is activated following 10 min of Rapamycin treatment and remained upregulated at 60 min (Figure 4B). Total p90-RSK1 remained unaffected at early time points (Figure 4B). PANC-1 cells show a loss of 4EBP1 protein compared to the MiaPaca2 cells (Appendix A). Next, we compared the effect of Rapamycin on signaling in MiaPaca-2 cells. MiaPaca-2 cells showed a similar extent of inhibition of phospho-S6 and feedback activation of phospho-AKT1 within 60 min (Figure 4C). However, unlike PANC-1, MiaPaca-2 did not show significant changes in phospho-eIF4E and phospho-p90-RSK1 (Figure 4D). Phospho-S6 remains inhibited at longer time points in PANC-1 while phospho-AKT1 remained upregulated suggesting that S6-dependent translation remains inhibited in the presence of feedback activation of phospho-AKT1 (Appendix A). In addition, Rapamycin induced the translation of p90-RSK1 (RPS6KA1) in later time points (Figure 3G). p90-RSK1 (RPS6KA1) can further feed to activation of S6, eIF4B, and cap-dependent translation [26]. In summary, these data suggest that Rapamycin affects the translation of mRNAs in a specific manner in pancreatic cancer cells that lack 4EBP1 expression likely through phospho-AKT and/or phospho-p90-RSK1 activation. Rapamycin-mediated reduction of p70-S6K1 total protein and phospho-S6 results in inhibition of translation of cell cycle driving and cancer growth proteins. However, feedback activation of AKT1, eIF4E, and p90-RSK1 (RPS6KA1) contribute to the translational upregulation of proteins involved in PI3K-AKT-mTOR, TNFa, and IL2-STAT signaling (Figure 4E). Based on these, we tested the effect of the p70-S6K1 inhibitor in combination with rapamycin. p70-S6K1 inhibitor did not show any additional effect in combination with Rapamycin (Appendix A). Others have reported the translational effect of Rapamycin and other mTOR inhibitors in cells with functional/wildtype 4EBP1. In Thoreen et al., 2012 the authors performed ribosome footprinting with Torin 1 in wildtype (WT) and 4EBP1/2 double KO (KO) MEFs [10]. In 4EBP1 wild-type cells, they identified 1872 genes in the TE down and 2968 genes in the TE up category. Similarly, in the 4EBP1/2 double KO, they found 1833 genes in TE down and 3007 genes in TE up groups (Appendix A). In both WT and KO cells, they observe a higher number of genes in TE up compared to the TE down subset suggesting that feedback activation of the translation may not depend on the 4EBP1 status (Appendix A). Concurrently, we observe that the phospho-AKT1 activation is present in both PANC-1 and MiaPaca2 cells and may be independent of the 4EBP1 status (Figure 4A,C). In another study reported by Hseih et al., 2012 the authors performed ribosome footprinting following Rapamycin treatment in PC3 prostate cancer cells [22]. There was no significant difference in the number of TE up and TE down genes (Appendix A). Hseih et al., 2012 also used the allosteric inhibitor of mTOR, PP242 which reduced the p-AKT, p-S6, and p-4EBP1. Surprisingly with the PP242, they did not observe any genes showing significant TE upregulation (Appendix A). Hseih et al., 2012 report that in PC3 prostate cancer cells rapamycin only reduced the p-S6 and not the phospho-AKT and phospho-4EBP1. Together, the presented data indicates that phospho-AKT may be required for the feedback translational upregulation. Next, we utilized the luciferase translation reporter assay driven by RNA-G quadruplex reported in our previous study [27] to identify the effect of Rapamycin on eIF4A-dependent translation. We observed that Rapamycin treatment slightly enhanced the RNA G-quadruplex mediated translation in PANC1 cells (Figure 4F). Accordingly, eIF4A inhibitor CR-1-31B enhanced the anti-proliferative effect of rapamycin in PANC-1 and MiaPaca2 cells (Figure 4G and Appendix A). We calculated the synergy score using Synergy Finder. In Panc1, the *p*-values for synergy scores were not significant (*p* > 0.05), but with ZIP and Bliss scoring system the drugs were found to likely be synergistic, while with Loewe and HSA they were found to likely be additive. In Miapaca2 cells, the synergy scores show the drugs to likely be additive, with three scoring systems (ZIP, HSA, Bliss), and showed significant *p*-values. The combined sensitivity score of both drugs was not higher than just CR-1-31B alone in both cell lines suggesting that inhibiting eIF4A alone may elicit a greater response than inhibiting Rapamycin alone. Synergy scores and drug interactions are shown in Figure 4H. Both PANC-1 and MiaPaca2 cells showed comparable responses to CR-1-31B alone suggesting that the status of 4EBP1 did not affect the sensitivity to the eIF4A inhibitor (Appendix A). In summary, we show that mTOR inhibition results in the feedback activation of translation programs in 4EBP1 lacking pancreatic cancer cells, and targeting translation downstream of mTOR may be a better therapeutic strategy in pancreatic cancer.

## 3. Discussion

mTOR controls protein synthesis and supports cancer growth by activating S6 and 4EBP1/eIF4E/eIF4A-dependent translation programs [3,4,21,27,28,29,30]. Oncogenic signaling pathways such as PI3K, KRAS, and MYC converge at mTOR to activate the oncogenic translation program; hence, mTOR inhibition has been the target of many therapeutic studies [14,31,32]. However, pancreatic cancer cells bypass mTOR inhibition through loss of 4EBP1 expression, suggesting that S6 may have a differential role in mRNA translation [17,33]. Rapamycin, a well-known mTOR inhibitor, affects the translation of mRNAs in a cell type-specific manner through its alternate effects on phospho-S6 and 4EBP1 signaling [34]. In this study, we established that rapamycin treatment directly affects translation in pancreatic cancer cells PANC-1, lacking 4EBP1 expression. Rapamycin inhibited the translation of cell cycle and cancer growth-promoting genes such as p70-S6K, explaining the anti-proliferative effect of rapamycin on pancreatic cancer cells.

However, rapamycin also induced the translation of a larger subset of mRNAs which includes p90-RSK1 (RPS6KA1) and MKNK2 (MNK2), indicating that rapamycin may activate the eIF4E-eIF4A-dependent translation. Rapamycin also activated the phospho-AKT1 while the phospho-S6 remains inhibited suggesting that the feedback activation of AKT1 may feed to eIF4E and activate the translation. In our previously published study, we have characterized the translational changes dependent on the 4EBP1 by using a doxycycline-inducible 4EBP14A mutant form in 293 cells. We observed that 4EBP1 majorly regulates the translation of proteins involved in insulin signaling and glucose metabolism [30]. However, in this study, we observe a distinct subset of mRNAs being affected likely by phospho-S6 inhibition in pancreatic cancer cells suggesting a distinct role of S6-dependent translation. Previously we have shown that eIF4A regulated the translation of RNA G-quadruplex containing mRNAs, and this includes key oncogenes such as MYC and KRAS [27,35]. We established that the eIF4A inhibitor CR-1-31B shows therapeutic activity in MYC-driven leukemia and KRAS-driven pancreatic cancer [35]. Rapamycin inhibited the p70-S6K while activating the p90-RSK that can signal to eIF4E, and accordingly, we observed increased phosphorylation of eIF4E. Based on this we hypothesized that targeting eIF4A with CR-1-31B may enhance the anti-proliferative activity of rapamycin. While the p70-S6K inhibitor did not show a significant increase in cell growth inhibition, a combination of CR-1-31-B significantly enhanced the anti-proliferative effect of rapamycin in pancreatic cancer cells. However, the combination effect did not exceed the effect of CR-1-31B alone suggesting that targeting eIF4A downstream of mTOR may show a better therapeutic effect regardless of the 4EBP1 and phospho-AKT status.

In summary, we characterize the translational targets of rapamycin in 4EBP1 deficient pancreatic cancer cells and show that mTOR translationally controls the p70-S6K protein expression. In addition, we establish the feedback activation of translation of key cancer growth-promoting proteins such as p90-RSK1 and MKNK2 that signals to eIF4E-eIF4A dependent translation following mTOR inhibition. Accordingly, targeting eIF4E-eIF4A-dependent translation downstream of mTOR enhances the activity of rapamycin and this may be an effective therapeutic strategy to target oncogenic translation programs in cancer.

## 4. Materials and Methods

### 4.1. Cell Culture and Treatments

Cancer cell lines were obtained from American Type Culture Collection and cultured as per instructions. PANC-1 cells were cultured in DMEM (Gibco; Thermo Fisher Scientific, Inc.; Waltham, MA, USA) supplemented with 10% fetal bovine serum (FBS; MilliporeSigma; St. Louis, MO, USA) and 100 U/mL penicillin, 100 µg/mL streptomycin, 0.292 mg/mL glutamine (Gibco; Thermo Fisher Scientific, Inc.; Waltham, MA, USA). PANC10.05 cells were cultured in RPMI-1640 (Gibco; Thermo Fisher Scientific, Inc.; Waltham, MA, USA) supplemented with 10% fetal bovine serum and penicillin-streptomycin-glutamine. MiaPaca-2 cells were cultured in RPMI-1640 supplemented with 15% fetal bovine serum and penicillin-streptomycin-glutamine. Cells were treated with indicated drugs for indicated time points in complete media. Rapamycin, S6Ki, and [±]CR-1-31B were purchased from SelleckChem and MedChem.

### 4.2. Ribosome Footprinting

PANC-1 cells were treated with DMSO or Rapamycin (50 nM; 1 h). Total RNA and ribosome-protected fragments were isolated as per the published protocol [36]. Deep sequencing libraries were generated from these fragments and sequenced on the HiSeq2000 platform. Genome annotation was conducted with the human genome sequence GRCh37 downloaded from the Ensembl public database http://www.ensembl.org (accessed on 6 April 2016).

### 4.3. Sequence Alignment

Sequence alignment was carried out as described in our previous study [35]. Ribosome footprint (RF) reads were filtered such that only reads with a minimum quality score of 25 were kept for at least 75% of nucleotides. The linker sequence was trimmed from the 3′ ends of the reads. Reads shorter than 15 nt were filtered out. These pre-processing steps were conducted with FASTX-Toolkit (http://hannonlab.cshl.edu/fastx_toolkit/index.html, accessed on 4 June 2016). Removal of ribosomal RNA was conducted by aligning RF reads to the ribosomal RNA sequences of GRCh37 downloaded from UCSC Table Browser (https://genome.ucsc.edu/cgi-bin/hgTables, accessed on 4 June 2016). The reads were then mapped to GRCh37 using HISAT2. (http://daehwankimlab.github.io/hisat2/, accessed on 4 June 2016) with default parameters. Only uniquely aligned reads were used for further analysis. Similar to RF reads, total mRNA sequencing reads were similarly aligned to GRCh37 using HISAT2, and splice alignment for paired-end mRNA-seq datasets was performed with default parameters. Only uniquely aligned reads were used for further analysis. Alignment quantification of both RF and mRNA sequencing was conducted with featureCounts [37], using the annotations of GRCh37 protein-coding genes as input. For further analysis, we only used reads aligned to exonic regions of the protein-coding genes.

### 4.4. Footprint Profile Analysis

Ribo-Diff [38] was used to analyze translation efficiency based on ribosome footprinting and mRNA sequencing data. Genes with significantly changed translation efficiency were defined by a q-value cut-off of 0.05.

### 4.5. Metagene Analysis

The metagene2 package in R was used to construct metagene plots of RF coverage. To normalize coverage by the length of each subregion, binned RF coverage of Rapamycin and DMSO samples were computed for the 5′ UTR (10 bins), CDS (20 bins), and 3′ UTR (10 bins) subregions. RF coverage subregions were concatenated, averaged across replicate samples, and normalized to the maximum value in each plot. The Cufflinks package was used to select the predominant isoform for each gene [39,40].

### 4.6. Global mRNA Translation

PANC-1 cells were labeled for protein synthesis using Click-iT^®^ AHA metabolic labeling reagent as per the manufacturer’s instructions (Invitrogen; Thermo Fisher Scientific, Inc.; Waltham, MA, USA). Briefly, cells were treated with either rapamycin (for 30 min, 1 h, 2 h, or 4 h) or DMSO. Cells were incubated in a methionine-free medium for 30 min before AHA labeling for 1 h. Then, they were fixed with 4% paraformaldehyde in PBS for 15 min, permeabilized with 0.25% Triton X-100 in PBS for 15 min, and then washed with 3% BSA. Cells were stained using Alexa Fluor 488 Alkyne with the Click-iT Cell Reaction Buffer Kit (both Invitrogen; Thermo Fisher Scientific, Inc.; Waltham, MA, USA). Flow cytometry was used to detect changes in mean fluorescence intensity.

### 4.7. Cell Viability Assay

To generate IC_50_ curves, cells were treated with rapamycin from varying concentrations of 0.5 nM to 50 µM or a combination of rapamycin and either S6Ki (100 nM) or [±]CR-1–31B (10 nM) for 72 h. Cell viability was measured using ATP quantification with the CellTiter-Glo Luminescent Cell Viability Assay (Promega; Madison, WI, USA). The drug combination effect was calculated by using the Synergy Finder R package [41]. Briefly, the overview of synergy scores quantifies the effects of combining rapamycin with CR-1-31B in PANC-1 and MiaPaca2 cells. A score lower than −10 can be interpreted as antagonistic, between −10 and 10 as an additive, and higher than 10 as synergistic. We used four different models to calculate an expected response. The ZIP model references an additive effect as though the drugs do not interact; the Bliss model references the probability of additive effect between the drugs as though they are independent events; the Loewe model references an expected response as though the two drugs were identical; and the HSA model references the highest single drug response [41].

### 4.8. Clonogenic Survival Assay

Cells were seeded in 6-well plates (25 × 10^3^ cells/well for PANC-1 and 50 × 10^3^ cells/well for PANC10.05) and allowed to adhere overnight in regular media. DMSO or Rapamycin (0.1 µM, 0.5 µM, or 1.0 µM) was added and refreshed every 3 days until the end of the experiment at day 10. Treated cells were fixed in 10% formalin solution and stained with 0.1% crystal violet before images were taken. For quantification, we used the ImageJ software 1.53v. We set the color threshold to include visible colonies within each well. The Analyze Particles feature was used to measure the percentage of surface area covered by cells.

### 4.9. Immunoblotting

Lysates were made using lysis buffer consisting of 50 mM Tris HCl at 7.5 pH, 250 mM NaCl, 0.5% (*v/v*) NP-40, and 5 mM EDTA, with proteases and phosphatases. A total of 50 µg of protein was loaded into SDS-PAGSE gels and transferred onto iBlot 2 nitrocellulose membranes (Invitrogen). The following antibodies were purchased from Cell Signaling Technology: RPSKB1, RPSKB2, RPSKA1, STAT5A, AKT1, p-AKT S473, p-S6, S6, p-eIF4E S209, and eIF4E. β-actin (A5316) was purchased from Sigma.

### 4.10. Luciferase Reporter Assay

We used the RNA G-quadruplex-driven luciferase reporter assay as described in our previous study [27]. The TOP motif luciferase reporter construct was obtained from Addgene (plasmid cat. number # 26611). PANC-1 cells were transfected with the luciferase plasmid and treated with Rapamycin (50 nM) for 24 h. Luciferase assays were performed using the Dual-Luciferase Reporter Assay System (Promega E1960, Madison, WI, USA) following the manufacturer’s instructions.

### 4.11. Statistical Analysis

A hypergeometric test was performed to test for the significance of the enrichment of the gene overlap in the KEGG pathway. All data were analyzed with two-tailed *t*-tests unless specified.

### 4.12. Online Content

Supplementary display items are available in the online version of the paper. Raw and processed data for the ribosome footprinting and total mRNA sequencing are deposited in the NCBI Gene Expression Omnibus database (awaiting GSE accession number).

## 5. Conclusions

Our study establishes the effect of mTOR inhibitor Rapamycin on translation programs in pancreatic cancer lacking 4EBP1 expression. We show that rapamycin induces feedback activation of eIF4A-eIF4E-dependent translation in cells lacking 4EBP1 protein to facilitate cancer cell growth and this can be further inhibited by using eIF4A specific inhibitor. 

## Figures and Tables

**Figure 1 cancers-15-01444-f001:**
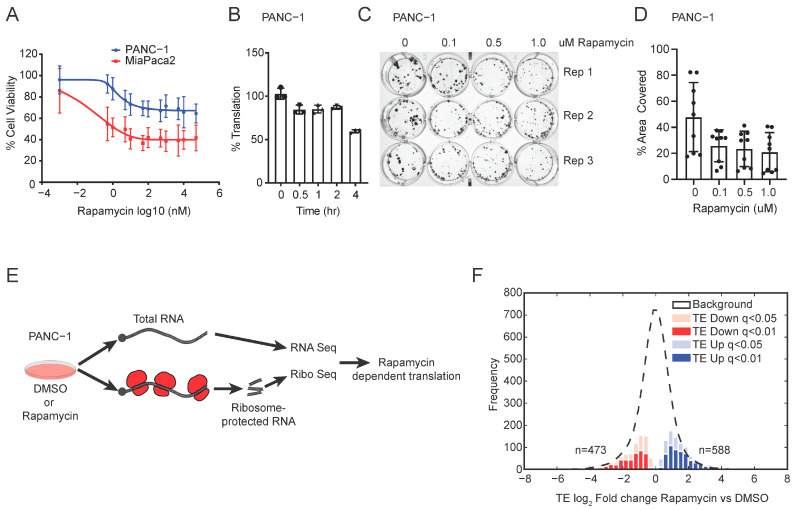
Ribosome footprinting identifies translational targets of mTOR and RPS6. (**A**) Cell viability assay showing the IC50 for rapamycin in PANC-1 and MiaPaca2 cells. (**B**) Global mRNA translation as measured by AHA incorporation in PANC-1 cells. (**C**,**D**) The clonogenic assay shows that rapamycin treatment inhibits cell growth in PANC-1 cells. (**E**) Schematic showing the experimental design for RNA seq and ribosome footprinting in PANC-1 cells treated with DMSO or rapamycin (50 nM; 1 h). Comparison of ribosome-protected sequences and total mRNA isolates the translational efficiency for each mRNA. (**F**) Frequency distribution of the change in translation efficiency (TE) in DMSO and rapamycin-treated PANC-1 cells. Using the statistical cut-offs of q < 0.01 and q < 0.05, mRNAs with decreased (TE down, red, and pink), increased (TE up, blue, and light blue), and unchanged (background, in dashed line) mRNA transcripts were identified.

**Figure 2 cancers-15-01444-f002:**
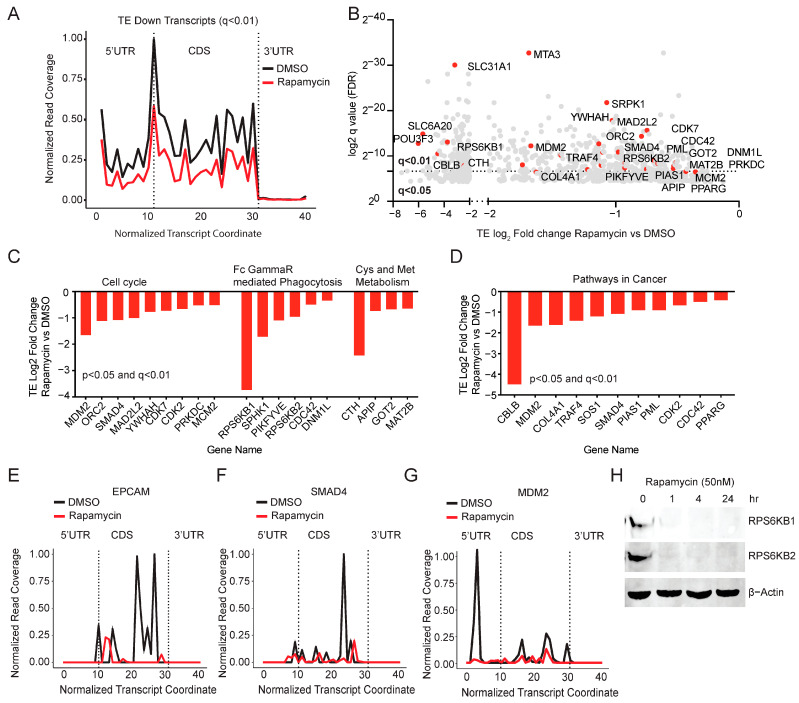
Rapamycin inhibits the translation of genes involved in the cell cycle and cancer cell growth. (**A**) Metagene plot showing a reduction in ribosome read coverage throughout the mRNA length in the TE down mRNAs (q < 0.01) in rapamycin-treated PANC-1 cells compared to the DMSO. RF coverage and transcript length are normalized for comparison. (**B**) TE down genes ranked by q value. Translational targets involved in the cell cycle and cancer growth are highlighted in red. (**C**,**D**) TE (log2-fold change) of key genes involved in cell cycle, Fc gamma receptor-mediated phagocytosis, Cys and Met metabolism, and pathways in cancer. (**E**–**G**) Ribosome coverage is reduced throughout the mRNA length in EPCAM, SMAD4, and MDM2 in rapamycin-treated PANC-1 cells compared to the DMSO. RF coverage and transcript length are normalized. (**H**) Immunoblot showed a reduction in the total protein of RPS6KB1 and RPS6KB2 following rapamycin treatment (50 nM) for the indicated time points. b-actin is used as the loading control.

**Figure 3 cancers-15-01444-f003:**
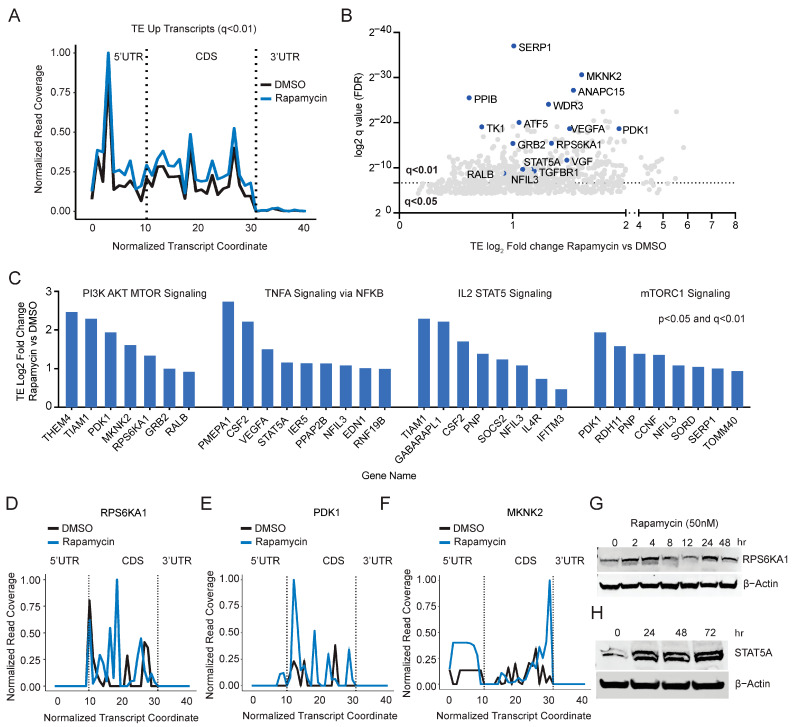
Rapamycin induces the translation of a subset of mRNAs. (**A**) Metagene plot showing that the ribosome read coverage is increased throughout the mRNA length in the TE up mRNAs (q < 0.01) in rapamycin-treated PANC-1 cells compared to the DMSO. RF coverage and transcript length are normalized. (**B**) TE up genes ranked by q-value. Translationally upregulated genes involved in PI3K-AKT-mTOR and cancer signaling are highlighted in blue. (**C**) TE (log2-fold change) of key genes involved in PI3K-AKT-mTOR, TNFA, IL2-STAT5, and mTORC1 signaling. (**D**–**F**) Ribosome coverage is increased throughout the mRNA length in RPS6KA1, PDK1, and MKNK2 in rapamycin-treated PANC-1 cells compared to the DMSO. RF coverage and transcript length are normalized for comparison. (**G**,**H**) Immunoblots showing upregulation in the total protein of RPS6KA1 and STAT5A following rapamycin treatment (50 nM) for the indicated time points. b-actin is used as the loading control.

**Figure 4 cancers-15-01444-f004:**
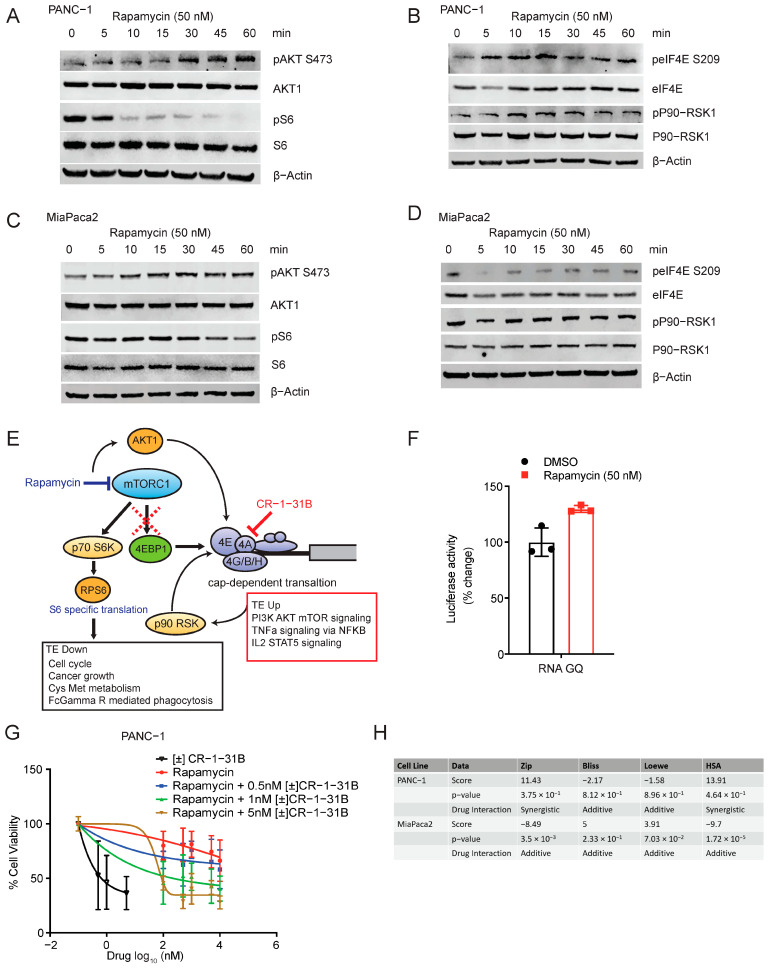
Rapamycin activates translation through feedback activation of AKT1 and eIF4E. (**A**) Immunoblot showing upregulation in the phospho-AKT1 and downregulation in phospho-S6 following rapamycin treatment (50 nM) for the indicated time points in PANC-1 cells. Total AKT1 and total S6 remain unaffected. β-actin is used as the loading control. (**B**) Immunoblot showing upregulation in the phospho-eIF4E and phospho-p90-RSK1 following rapamycin treatment (50 nM) for the indicated time points in PANC-1 cells. Total eIF4E and p90-RSK1 remain unaffected. β-actin is used as the loading control. (**C**) Immunoblot showing upregulation in the phospho-AKT1 and downregulation in phospho-S6 following rapamycin treatment (50 nM) for the indicated time points in MiaPaca2 cells. Total AKT1 and total S6 remain unaffected. β-actin is used as the loading control. (**D**) Immunoblot showed a slight reduction in the phospho-eIF4E and no change in phospho-p90-RSK1 following rapamycin treatment (50 nM) for the indicated time points in MiaPaca2 cells. Total eIF4E and p90-RSK1 remain unaffected. β-actin is used as the loading control. (**E**) Proposed model showing the effect of rapamycin on signaling and translation. Rapamycin inhibits phospho-S6 and the translation of genes involved in the cell cycle and cancer cell growth. Rapamycin upregulates the translation of RPS6KA1 (p90-RSK1) and activates signaling via phospho-AKT1 which feeds to eIF4E-eIF4A dependent translation. Rapamycin treatment activates the translation of genes involved in PI3K-AKT-mTOR, TNFa, and Il2-STAT5 signaling. Targeting eIF4A with inhibitors, such as CR-1-31B, downstream of mTOR may be an effective therapeutic. (**F**) Luciferase assay showing upregulation of RNA G-quadruplex (RNA GQ) dependent translation following Rapamycin treatment (50 nM; 24 h) in PANC-1 cells. (**G**) Cell viability assay shows that combination with CR-1-31B enhances the anti-proliferative effect of rapamycin in PANC-1 cells. (**H**) Synergy scores for Rapamycin and CR-1-31B combination in PANC-1 and MiaPaca2 shows additive/synergistic effect using various synergy score programs.

## Data Availability

Sequencing data generated in this study are available in the NCBI Gene Expression Omnibus database (GSE accession number GSE225683).

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
