# Peer review of "Rapamycin-Induced Feedback Activation of eIF4E-EIF4A Dependent mRNA Translation in Pancreatic Cancer"

_cancers, 2023, doi:10.3390/cancers15051444_

Round 1
Reviewer 1 Report (New Reviewer)
The Authors have developed a very interesting point of view of target therapy within the pancreatic tumor pathology. The authors used three different PDAC cell lines in their investigations. This trait is in accordance with journal policy. The experiments were well designed and the conceptual design of the paper is clear. The conclusion is supported by the evidence of the results.
The article is ready for publication in this form.
This manuscript is a resubmission of an earlier submission. The following is a list of the peer review reports and author responses from that submission.
Round 1
Reviewer 1 Report
Thank you very much for the opportunity to review this manuscript.
This reviewer thinks that some major revisions are needed:
- Result 2.1, lines 64-65: the authors declare that the three cell lines they chose to use in this project show growth inhibition following rapamycin treatment for three days. However, the cell viability assays showed do not really support this statement. MiaPaca2 seems to be the only one reaching an IC50, although the response to the treatment doesn’t seem to be [drug]-dependent. Please discuss or improve
- Figure 1 B and 1 c are wrongly labeled
- Fig 1B: is missing the cell name label
- Fig 1C: it would be clearer to the reader if the results of the clonogenic assay are shown as bar graph. The replicates can be put as supplementary. Moreover, the quality of the figure is very low
- Supplementary Fig1C: the results are not consistent with the data from Supp Fig 1B
- Fig 1D: the authors used rapamycin at 50nM concentration. How was this concentration chosen?
- Fig. 2B: please check the x axis
- When discussing the TE up results, the authors showed the KEGG pathways, while in the TE down they focused on Hallmark pathways. Why this change?
- Supp Fig 3E-3F: these are not discussed in the manuscript
- Fig 4A, Supp Fig4A: the concentration of rapamycin used is different. Why?
- Please, show pAKT S473 at later time points in Supp Fig4A
- Fig 4B: data about p90-RSK1 are missing
Reviewer 2 Report
The authors employ pancreatic cancer cells that lack the expression of 4E-BP1 to perform ribosomal footprint analysis to identify mRNA populations that exhibit either repressed or enhanced translation upon rapamycin treatment. Based on the results, the authors suggest that lack of 4E-BP1 causes inability of rapamycin to efficiently inhibit eIF4E-dependent translation and suggest that use of inhibitors downstream of mTOR can potentially compensate for that. I find this study technically sound as it successfully combines genome-wide analyses with biochemical experiments. There are several points that I would suggest strengthening, as detailed below.
1. Figures 1B and C are confused in the text of 2.1.
2. In 2.1 the authors identify mRNAs that demonstrate either repressed or enhanced translation. Later in the text, the authors claim that enhanced translation is due to reduced ability to inhibit eIF4E because 4E-BP is missing in this particular cells. It can help if the authors would present some ribosomal profiling data from cells with intact 4E-BP after rapamycin treatment. For that, they can analyze publicly available datasets. Would these cells exhibit less mRNAs with upregulated translation? While this might be an obvious question for the experts in the field, showing these results may help the general readers to see the differences the authors refer to.
3. It is a bit strange that the authors do not particularly test the translation (or expression level) of the ribosomal proteins that have 5’TOP sequences. Active eIF4E should prevent the inhibition of their translation, so showing their ribosomal profiles and/or WB can strengthen the authors’ point.
4. Line 170: the authors write that Figure 4B shows the expression of p90. But this panel shows the expression of eIF4E instead.
5. The authors also suggest that the inability of Rapamycin to demonstrate a major inhibition effect stems from the absence of 4E-BP in these cells. The authors could be right, particularly based on the findings of Choo et al., PNAS 2008, but this conclusion appears to be insufficiently supported by their data. Is it possible to transfect the cells lacking 4E-BP with plasmids encoding for this gene and show that the complementation of this protein restores rapamycin-mediated inhibition? This could be done on a few genes using WB analysis. The authors could potentially employ a mutant 4E-BP version that cannot be phosphorylated, which can make their point even stronger. Do the authors think such an experiment could be feasible?
6. In the Figure 4D the authors show the effect of combined treatment, which exceeds the impact of rapamycin. However, this effect can be solely due to inhibition of eIF4A, and the authors should compare the combined effect to the only eIF4A inhibition to prove their point.